# Dynamics-Inspired Text-Guided Video Storytelling

## Abstract

Generating coherent long-form video sequences from discrete input using only text prompts is a critical task in content creation. While diffusion-based models excel at short video synthesis, long-form storytelling from text remains largely unexplored and a challenge due to difficulties in temporal coherency, preserving semantic meaning, and maintaining both scene context and action continuity across the video. We introduce a novel storytelling framework that achieves this by integrating scene and action prompts through dynamics-inspired prompt mixing. Specifically, we first present a bidirectional time-weighted latent blending strategy to ensure temporal consistency between segments of the long-form video being generated. We then propose a dynamics-informed prompt weighting (DIPW) mechanism that adaptively balances the influence of scene and action prompts at each diffusion timestep by jointly considering CLIP-based alignment, narrative continuity, and temporal smoothness. To further enhance motion continuity, we incorporate a semantic action representation to encode high-level action semantics into the blending process, dynamically adjusting transitions based on action similarity and ensuring smooth yet adaptable motion changes. Latent space blending maintains spatial coherence between objects in a scene, while time-weighted blending enforces bidirectional constraints for temporal consistency. The resulting integrative system prevents abrupt transitions while ensuring fluid storytelling that faithfully reflects both scene and action cues. Extensive experiments demonstrate significant improvements over baselines, achieving temporally consistent and visually compelling video narratives without any additional training. This approach bridges the gap between short clips and extended video to establish a new paradigm in GenAI-driven video synthesis from text.

## 1 Introduction

Storytelling through text-based video synthesis presents a challenge in content creation Li et al. (2019); Maharana et al. (2021); Zhuang et al. (2024); He et al. (2023; 2024); Zhao et al. (2024); Wang et al. (2024b); Zheng & Fu (2024). Recent advances in diffusion-based models have significantly improved the quality of short video generation. However, these models often struggle to generate long-form coherent video sequences. When videos are extended beyond a few seconds, noticeable inconsistencies often appear, disrupting the overall flow AI (2024); Yin et al. (2023); Qiu et al. (2023); Wang et al. (2023a); Oh et al. (2024); Zheng & Fu (2024); Villegas et al. (2022).

One approach to generate longer videos for storytelling is to create several short clips, each guided by a distinct text prompt that narrates different segments of the story Zhuang et al. (2024); He et al. (2023; 2024); Zhao et al. (2024); Wang et al. (2024b). While this method shows promise, it still encounters significant issues with maintaining consistency across the various clips. The primary limitation lies in the inability to establish *temporal coherence* across multiple short clips, while *preserving semantic meaning and action continuity*.

Temporal coherence is vital to ensure that visual elements, characters, and actions in a video remain consistent from one clip to the next. Without it, viewers may notice jarring transitions, discrepancies in character appearance or behavior, and interruptions in the storyline. Achieving temporal coherence requires a sophisticated understanding of the narrative context and the ability to seamlessly integrate each segment into a unified whole Chen et al. (2024); Kim et al. (2019); Chu et al. (2020); Niklaus et al. (2017); Srivastava et al. (2015). Additionally, preserving semantic meaning and action

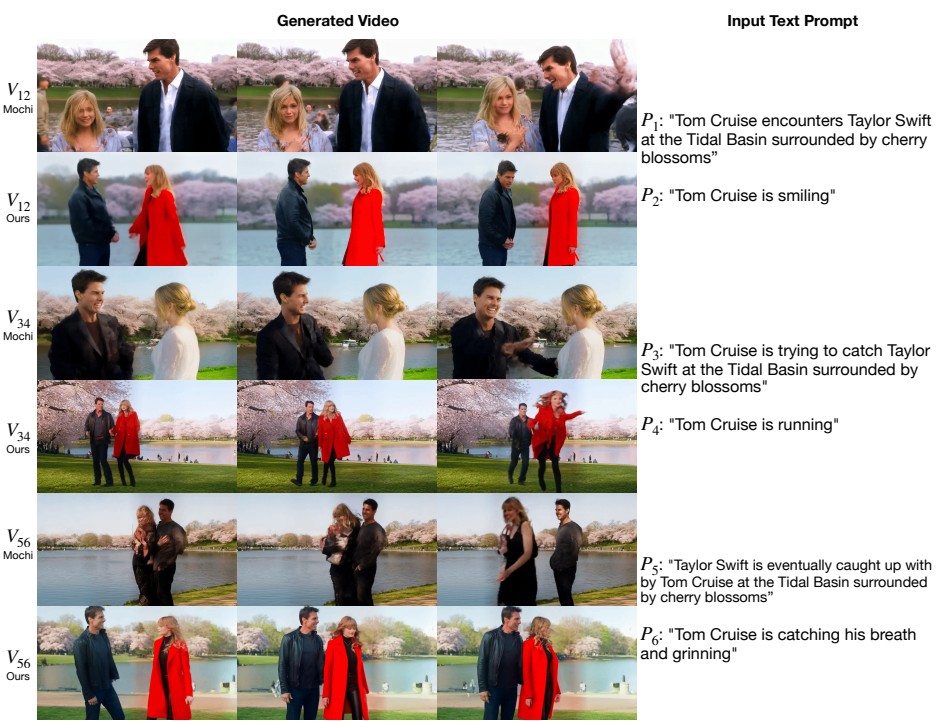

Figure 1: **Example of story generation using our proposed method** Each row represents a video segment $V_{12}$, $V_{34}$, $V_{56}$, where the scene and motion evolve according to the specified textual prompts. The first prompt ($P_1$) describes a detailed action (e.g., "Tom Cruise is trying to catch Taylor Swift at the Tidal Basin surrounded by cherry blossoms"), with the second prompt ($P_2$) follows (e.g., "Tom Cruise is running"). Our approach ensures smooth transitions between these states using Time-weighted Blending (TWB), Dynamics-Informed Prompt Weighting (DIPW), and Semantic Action Representation (SAR), preserving consistency across entire story videos. Our method (Bottom, Even Row) provides much more consistent scenes compared to the text-to-video generation model, Mochi baseline (Top, Odd Row), enabling smooth video storytelling.

continuity implies that each clip must not only align visually but also follow the logical progression of the narrative. This involves maintaining character arcs, ensuring that actions initiated in one clip are carried through in subsequent clips, and retaining the overall thematic and emotional tone of the story. DreamRunner Wang et al. (2024b) introduces retrieval-augmented test-time adaptation and spatial-temporal 3D attention for object-motion binding. Despite its claimed improvements, it fails to ensure scene-to-scene consistency. Transitions between clips are often abrupt, and character appearance and motion lack continuity, disrupting the overall storytelling experience.

We present a novel approach for storytelling by overcoming the above challenges to enable seamless transitions between generated video segments, thereby creating cohesive, longer-length stories from discrete prompts. Our approach contains three methodological contributions that address various challenges in current video storytelling works:

1. **Dynamics-Informed Prompt Weighting (DIPW).** While most prior video storytelling frameworks rely on a single prompt, this setting often struggles to capture both the *thematic richness of a scene* and the *dynamic details of actions*. To address this limitation, we adopt a dual-prompt formulation that explicitly separates scene description from action specification. We introduce *Dynamics-Informed Prompt Weighting (DIPW)*, which adaptively balances the contributions of these two prompts throughout the denoising process. This design allows the model to integrate emotional subtleties, thematic depth, and action continuity within each scene, resulting in more expressive and coherent video narratives.

2. **Time-Aware Blending with Bidirectional Constraints.** To prevent abrupt visual or motion discontinuities across frames, we propose a *Time-Weighted Blending (TWB)* strategy

with bidirectional constraints. TWB balances information from both past and future latent states, enforcing smooth appearance and motion evolution. Frame continuity constraints and transition smoothing further refine temporal progression, ensuring seamless video transitions.

3. **Structured Semantic Action Representation.** Even when temporal consistency is preserved, action sequences may drift semantically. To address this, we introduce *Structured Semantic Action Representation (SAR)*, which encodes high-level action semantics with a pre-trained text encoder (e.g., CLIP). By embedding semantic similarity into the blending process, SAR preserves logical action progressions and enhances narrative coherence across video segments.

Our framework unified these *dynamics-inspired* components to create extended sequences that maintain temporal-spatial coherence and exhibit logical, progressive storytelling. By iteratively applying structured blending techniques across multiple short video segments, our approach enforces consistency in motion dynamics and in video sequences seamlessly, thereby contributing to a cohesive narrative.

Extensive qualitative and quantitative analysis and ablation studies demonstrate the effectiveness of our approach in generating long-form video narratives that maintain temporal coherence and semantic consistency. We showcase **our model that outperforms prior methods with the highest CLIP-add (+3.91% vs. Mochi, +5.74% vs. Vlogger), CLIP-combined (+3.49%, +7.10%), and DINO (+1.75%, +4.78%) scores**, demonstrating superior text alignment and coherence. By bridging the gap between isolated video segments and extended storytelling, our work paves the way for more advanced and realistic GenAI-driven video synthesis applications.

## 2 RELATED WORKS

### 2.1 STORYTELLING VIDEO GENERATION

Storytelling video generation aims to create long, multi-scene videos that remain consistent with the story described in a given script. VideoDirectorGPT Lin et al. (2023) and Vlogger Zhuang et al. (2024) leverage large language models (LLMs) for high-level script decomposition, structuring multi-scene conditions to generate videos sequentially. Animate-A-Story He et al. (2023) improves motion control by retrieving depth-conditioned reference videos, ensuring more accurate motion representation. More recently, DreamStory He et al. (2024) and MovieDreamer Zhao et al. (2024) employ text-to-image models to generate keyframes, which are then animated using image-to-video models to maintain temporal coherence. Additionally, customization techniques Wang et al. (2024b); He et al. (2024); Zhao et al. (2024) have been explored to enhance character consistency across different scenes. However, these methods primarily rely on keyframe-based animation and may struggle with generating smooth, dynamic motion transitions. DreamRunner Wang et al. (2024b) incorporates retrieval-augmented adaptation to capture motion priors from reference videos, enabling more diverse and customizable scripted motions. It also introduces a spatial-temporal 3D attention module for improved motion binding. However, it relies on motion prior training and large-scale LLMs, increasing computational cost, and limiting scalability. Also, the consistency between videos is not maintained. StoryDiffusion Zhou et al. (2025) proposes Consistent Self-Attention, a self-attention modification that improves image consistency in pretrained diffusion-based models. It further introduces a Semantic Motion Predictor to estimate motion transitions in the semantic space, enhancing video smoothness. Despite these advancements, StoryDiffusion still requires training and struggles with generating complex scenes. Also, it primarily addresses image-to-video or comic generation rather than story-driven video synthesis from pure text prompts. Unlike existing approaches that rely on motion priors, keyframe animation, or retrieval-based adaptation, our method can achieve controlled motion evolution without requiring additional training or large-scale datasets. Our setting explicitly decomposes prompts into **scene descriptions and action commands**, enabling prompt mixing that simultaneously captures scene evolution and action continuity. This design choice reflects our focus on **story generation**: real narratives are not defined by static descriptions alone, but by how characters and objects interact within evolving scenes. By separating scene and action, our framework can model both the background progression and the foreground dynamics, which together form the backbone of coherent storytelling—an aspect largely absent in existing approaches and not directly measurable with standard video consistency benchmarks.

## 2.2 Text-to-Video Diffusion

There has been immense work on text to video generation Bar-Tal et al. (2024); Chen et al. (2023a); Fei et al. (2024); Girdhar et al. (2023); Khachatryan et al. (2023); Qing et al. (2024); Singer et al. (2022); Wang et al. (2023b); Weng et al. (2024); Zhang et al. (2024; 2023); Henschel et al. (2024); Qiu et al. (2023); OpenAI (2024); Blattmann et al. (2023b); Ge et al. (2023); Wang et al. (2024a); Yin et al. (2023); He et al. (2022); Blattmann et al. (2023a); Wang et al. (2023a); Chen et al. (2023b); Oh et al. (2024); Villegas et al. (2022); Polyak et al. (2025); Sharma et al. (2024); Veo-Team et al. (2024). Mochi 1 AI (2024) leverages a diffusion-based framework with multi-stage conditioning to enhance motion fidelity and temporal consistency but struggles with generating fine-grained scene details and maintaining narrative coherence over extended sequences. CogVideoX Yang et al. (2024), an extension of CogVideo Hong et al. (2022), employs a transformer-based architecture to produce high-resolution videos with long-range coherence, though it requires significant computational resources and still faces challenges in consistent storytelling. While both methods advance text-to-video generation, they are designed around single descriptive prompts and therefore have limitations in sustaining structured, multi-segment narratives.

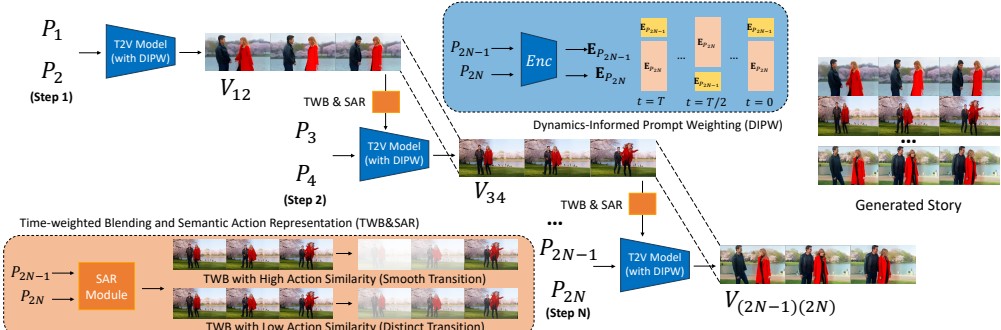

Figure 2: **Overview of our proposed storytelling framework.** Each video segment $V_{2N-1,2N}$ is generated from a pair of prompts $(P_{2N-1}, P_{2N})$, ensuring coherent short-form synthesis. These segments are then sequentially combined to form a complete story. We incorporate Time-Weighted Blending (TWB) and Semantic Action Representation (SAR) to maintain temporal coherence and logical action continuity across segments. TWB ensures smooth transitions by dynamically adjusting blending weights based on prior frames, while SAR refines transitions based on action similarity. The framework builds upon the Mochi model, enhanced with DIPW-based prompt weighting, ensuring structured motion evolution. Transitions between segments adaptively vary, with high action similarity producing smooth transitions and low action similarity allowing distinct but logical shifts in motion.

## 3 Method

**Problem Statement** Our goal is to generate coherent long-form video sequences from text, using pretrained text-to-video models, without any additional training data or finetuning. Specifically, the inputs to our model are a sequence of paired text prompts $(P_1, P_2), (P_3, P_4), ..., (P_{2N-1}, P_{2N})$, where each pair describes a segment of the story. The first prompt $P_{2N-1}$ provides background and context, while the second prompt $P_{2N}$ specifies character actions and movement. The desired output is a set of video segments $\{V_{12}, V_{34}, ..., V_{(2N-1)(2N)}\}$, creating a structured and coherent long-form video story, while ensuring smooth transitions and narrative consistency, where each video $V_{2N-1,2N}$ corresponds to the generated sequence from the paired prompts $(P_{2N-1}, P_{2N})$.

**Overview of approach.** Pretrained text-to-video models, while successful in generating short videos, often struggle with extended storytelling or generating long-form video sequences, and produce abrupt scene changes and motion inconsistencies. Our approach addresses the key challenges of temporal coherence, smooth transitions between distinct prompts, and maintaining logical action sequences by proposing three techniques: (1) Prompt Smoothing - We present a novel Dynamics-Informed Prompt Weighting (DIPW) framework to video generation, enabling structured and controlled prompt interpolation that guides the evolution of generated frames. (2) Time-Aware

Latent Space Blending - We incorporate latent space blending with time-weighted blending within a diffusion-based framework. This ensures smooth interpolation between video segments while preserving object consistency and action coherence. (3) Semantic Action Representation (SAR) - Our SAR mechanism refines the blending process by dynamically adjusting transitions based on action similarity, further enhancing motion continuity. Figure 2 illustrates our method. Our method first applies DIPW to dynamically select the most suitable prompt embedding at each denoising steps, influencing the conditioning of the diffusion process; then, Time-Weighted Blending ensures smooth transitions by incorporating decayed prior frames, and finally, Semantic Action Representation refines motion continuity by adjusting blending factors based on action similarity, resulting in a coherent long-form video.

## 3.1 DYNAMICS-INFORMED PROMPT WEIGHTING (DIPW)

Existing video storytelling methods often rely on a single prompt formulation. However, this design makes it difficult to simultaneously represent the *emotional and thematic aspects of a scene* together with the *detailed action dynamics* that drive narrative progression. To enable temporally coherent video generation conditioned on multiple discrete text prompts (e.g., scene-level and action-level descriptions), we introduce Dynamics-Informed Prompt Weighting (DIPW), a dynamic blending mechanism that adaptively determines the influence of each prompt during the diffusion denoising process. Unlike other strategies that treat prompt mixing, DIPW directly learns a time-dependent attention weight based on three key factors: (i) semantic alignment between frame and prompt, (ii) temporal smoothness with prior timestep embeddings, and (iii) narrative progression across the video.

At each denoising timestep $i \in [1, T]$, we compute the weights $\alpha_{2N-1}^i$ and $\alpha_{2N}^i$ for two prompts $P_{2N-1}$ and $P_{2N}$ as follows. Let:

- $\text{sim}_{2N-1}^i$: CLIP similarity between the generated frame and prompt $P_{2N-1}$,

- $\text{sim}_{2N}^i$: CLIP similarity with prompt $P_{2N}$,

- $\text{prev\_sim}_{2N-1}^i$: cosine similarity between current $P_{2N-1}$'s embedding and previous combined embedding,

- $\text{prev\_sim}_{2N}^i$: same for $P_{2N}$,

- $\text{prior}_{2N-1}^i = 1 - \frac{i}{T}$, $\text{prior}_{2N}^i = \frac{i}{T}$: narrative progression priors.

We define prompt scores as:

$$s_{2N-1}^i = \lambda_1 \cdot \text{sim}_{2N-1}^i + \lambda_2 \cdot \text{prev\_sim}_{2N-1}^i + \lambda_3 \cdot \text{prior}_{2N-1}^i,$$

$$s_{2N}^i = \lambda_1 \cdot \text{sim}_{2N}^i + \lambda_2 \cdot \text{prev\_sim}_{2N}^i + \lambda_3 \cdot \text{prior}_{2N}^i,$$

where $\lambda_1, \lambda_2, \lambda_3$ are fixed hyperparameters controlling the influence of alignment, smoothness, and prior, respectively.

To compute prompt weights, we apply a temperature-controlled softmax:

$$\tilde{s}_{2N-1}^i = \frac{s_{2N-1}^i - \max(s_{2N-1}^i, s_{2N}^i)}{\tau},$$

$$\tilde{s}_{2N}^i = \frac{s_{2N}^i - \max(s_{2N-1}^i, s_{2N}^i)}{\tau},$$

$$\alpha_{2N-1}^i = \frac{e^{\tilde{s}_{2N-1}^i}}{e^{\tilde{s}_{2N-1}^i} + e^{\tilde{s}_{2N}^i}}, \quad \alpha_{2N}^i = 1 - \alpha_{2N-1}^i.$$

Here, $\tau$ is a temperature parameter that controls the sharpness of the weighting; lower values encourage sharper selections, while higher values yield smoother blending.

The final prompt embedding used to condition the transformer at timestep $i$ is then computed as:

$$\mathbf{E}_i = \alpha_{2N-1}^i \cdot \mathbf{E}_{P_{2N-1}} + \alpha_{2N}^i \cdot \mathbf{E}_{P_{2N}},$$

where $\mathbf{E}_{P_{2N-1}}, \mathbf{E}_{P_{2N}}$ denote the encoder embeddings of the two prompts. The corresponding attention mask is selected based on the dominant weight to maintain alignment between the prompt semantics and the attention scope. Since each prompt may describe different visual entities or actions, using the attention mask associated with the dominant prompt ensures that the model focuses on the correct spatial regions during denoising. This formulation allows the model to dynamically focus on the most contextually relevant prompt at each step while ensuring smooth transitions in both semantic content and motion. Compared to static or heuristic prompt selection, DIPW introduces a principled, continuous blending strategy grounded in both frame-level similarity and temporal coherence.

## 3.2 Time-weighted Temporally-Aware Latent Space Blending (TWB)

Video segments generated using not only text-to-video models but also video story generation models often struggle with maintaining smooth transitions, resulting in incoherent motion. To address this issue, we introduce a structured approach to prompt integration within the diffusion process, ensuring generated motion sequences follow a coherent narrative. By interpreting high-level action descriptions as structured motion constraints, our method maintains logical action transitions, enabling long-form storytelling beyond independent short clips. We achieve this by applying time-weighted blending, which enforces a decayed influence of earlier frames on the starting frame of each new segment.

Let $z_i$ be the latent embedding at *frame index i* for a given segment $V_{(2N-1)(2N)}$. Rather than generating each segment in isolation, we compute a time-weighted blend of latent embeddings from the immediately preceding segment $V_{(2N-3)(2N-2)}$. Specifically, the initialization latent $\tilde{z}_{(2N-1)(2N),0}$ for segment $V_{(2N-1)(2N)}$ is computed as:

$$\tilde{z}_{(2N-1)(2N),0} = \sum_{i=0}^{N-1} \tilde{w}_i \cdot z_{(2N-3)(2N-2),i}, \tag{1}$$

where the time-decayed weights $w_i$ are computed by and normalized as $\tilde{w}_i$:

$$w_i = 0.9^{(N-i-1)}, \tilde{w}_i = \frac{w_i}{\sum_{j=0}^{N-1} w_j}. \tag{2}$$

We then update the first frame of $V_{(2N-1)(2N)}$ using a controlled blending factor $\gamma$. This is performed by:

$$z_{(2N-1)(2N),0} \leftarrow \gamma \cdot z_{(2N-3)(2N-2),T} + \gamma \cdot \tilde{z}_{(2N-1)(2N),0} + (1 - 2\gamma) \cdot z_{(2N-1)(2N),0}, \tag{3}$$

where $z_{(2N-3)(2N-2),T}$ is the final latent frame of the preceding segment. Note that each segment is generated independently, but the initialization of its first frame softly incorporates information from the prior segment to promote motion continuity. The symbol $\leftarrow$ is used to indicate an assignment (update) operation. The same formulation applies to subsequent segments. For example, the first frame of $V_{56}$ is computed as:

$$z_{56,0} \leftarrow \gamma \cdot z_{34,T} + \gamma \cdot \tilde{z}_{56,0} + (1 - 2\gamma) \cdot z_{56,0}. \tag{4}$$

Equations 3 and 4 are implemented as recursive assignment rules during the diffusion sampling process.

## 3.3 Semantic Action Representation for Motion-Aware Blending (SAR)

Preserving temporal smoothness alone is insufficient, as action sequences may still drift semantically, resulting in narratives that lack logical progression or consistency. To further enhance motion continuity and ensure smooth transitions between actions in the video from one frame to another, we introduce a Semantic Action Representation (SAR) mechanism. This mechanism encodes high-level action semantics into the blending process. Starting from $P_3$, each prompt $P_i$ is embedded into an action representation space:

$$\mathbf{a}_i = f_A(P_i) \in \mathbb{R}^d,$$
(5)

where $f_A$ is a pre-trained text encoder such as CLIP, and $\mathbf{a}_i$ is the normalized action embedding. We compute the cosine similarity between consecutive prompts to quantify semantic alignment:

$$S_A(P_{2N-1}, P_{2N}) = \frac{\mathbf{a}_{2N-1} \cdot \mathbf{a}_{2N}}{\|\mathbf{a}_{2N-1}\|\|\mathbf{a}_{2N}\|}.$$
(6)

Using this similarity score, the blending factor is modulated as:

$$\alpha' = \alpha \cdot (1 - S_A(P_{2N-1}, P_{2N})).$$
(7)

Here, the adjusted blending factor $\alpha'$ is designed to reduce blending when the actions are semantically similar, and to apply stronger blending when the prompts describe dissimilar actions. This adaptive formulation enables the model to maintain coherence within continuous actions while allowing for more distinct transitions between different motions. As a result, SAR improves the overall temporal consistency of action representation across adjacent video segments.

# 4 EXPERIMENTS AND RESULTS

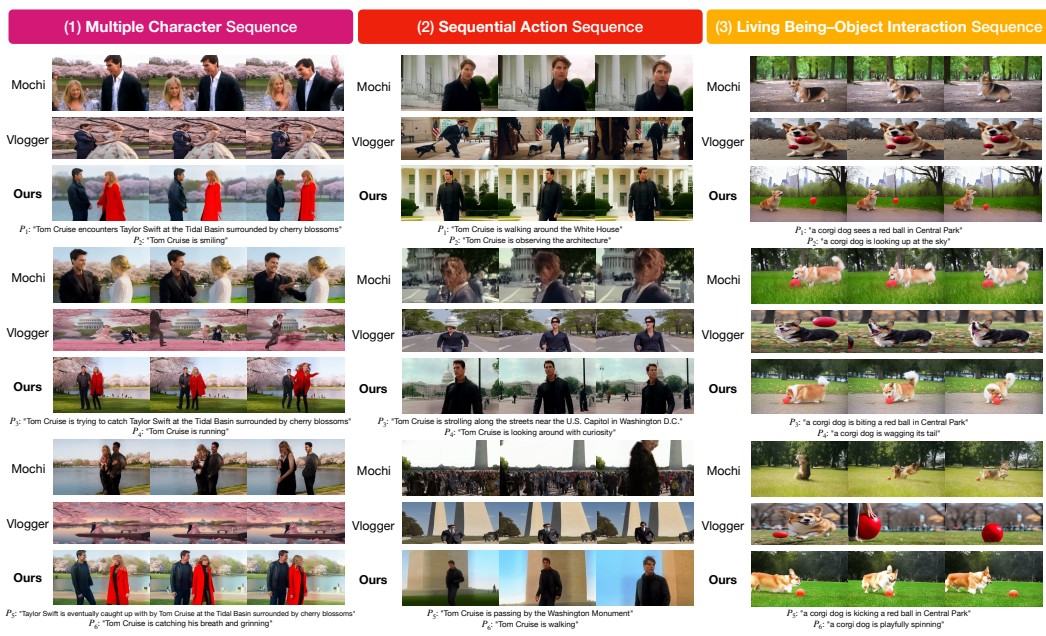

Figure 3: **Qualitative comparison across three sequence types:** (1) multiple character sequence, (2) sequential action sequence, and (3) living being–object interaction sequence. Our model consistently maintains character positioning, background consistency, motion continuity, and natural interactions across diverse scenarios. For additional video demonstrations, please refer to the supplementary material (see A.1).

## 4.1 IMPLEMENTATION DETAILS

We use the Mochi-1 AI (2024) backbone model as a T2V(Text-to-Video) model in all our experiments. The model operates in an inference-only setting, without any additional training. The video resolution is set to $848 \times 480$, and we use a classifier-free guidance scale Ho & Salimans (2022) of 4.5. The inference process involves 64 denoising steps per generation to ensure high-quality outputs. To enable prompt-level control during generation, we adopt the proposed Dynamics-Informed Prompt Weighting (DIPW) strategy. At each timestep, DIPW computes the blending weights of two prompt embeddings based on three components: (i) the CLIP similarity between the current

| Method | CLIP-add ↑ | CLIP-combined ↑ | BLIP ↑ | DINO ↑ | LPIPS $(V_{12} - V_{(2N-1)(2N)}) \downarrow$ |
|---|---|---|---|---|---|
| Vlogger | 0.2889 | 0.2998 | 28.8509 | 0.9254 | **0.6193** |
| Mochi | 0.2940 | 0.3103 | **29.5948** | 0.9530 | 0.6811 |
| **Ours (Full Model)** | **0.3055** | **0.3211** | 29.5117 | **0.9697** | 0.6733 |
| Ours w/o DIPW (no prompt weigthing) | 0.2883 | 0.3013 | 28.9450 | 0.9418 | 0.6412 |
| Ours w/o TWB (no temporal blending) | 0.2944 | 0.3030 | 29.1403 | 0.9655 | 0.6803 |
| Ours w/o SAR (no semantic action rep) | 0.2926 | 0.3005 | 28.7187 | 0.9526 | **0.6396** |

Table 1: **Quantitative evaluation of generated videos.** Higher CLIP and BLIP scores indicate better text alignment, while lower LPIPS values signify improved realism and story consistency. TWB indicates our temporally-aware latent space blending approach, SAR is the semantic action representation method and DIPW refers to the dynamics-informed prompt weighting (DIPW) based smoothing strategy. Our approach imparts significant benefits towards generating temporally consistent long-form videos, as also evidenced by the quantitative results.

frame and each prompt, (ii) temporal alignment with the previously applied embedding, and (iii) a diffusion-step-dependent prior. These components are equally weighted using fixed hyperparameters $\lambda_1 = \lambda_2 = \lambda_3 = 1.0$. The prompt with the higher weight determines the attention mask to maintain semantic focus and structural coherence. To avoid ambiguous weighting and encourage decisive transitions, the final weights are softly normalized such that one prompt serves as the dominant conditioning source while the other provides minimal but stabilizing influence. This weighting behavior was found to support both semantic alignment and motion continuity across long-form video segments. In our experiments, we use a fixed temperature of $\tau = 0.5$ to balance selectivity and stability during prompt interpolation. All experiments (see Appendix document) are conducted on a single H100 GPU (80GB).

## 4.2 DATASET

Coherent *story generation* requires not only visual consistency within a scene but also continuity across actions that drive the narrative forward. Unlike single-prompt video generation, where one caption is sufficient, storytelling inherently demands the interplay between *scene descriptions* (to ground the environment and context) and *action commands* (to advance the plot). However, no existing dataset directly supports this structured prompt-pair setting for long-form video storytelling. Inspired by Set 4 of Kothandaraman et al. (2024) for prompt image-based mixing, we construct a diverse data set for our task that includes both animate objects (e.g. humans, animals) and inanimate objects (e.g. vehicles, sports equipment). The dataset spans eight locations, with seven featuring multi-character interactions. Each story consists of 12-13 sequential scenes, and we introduce two character sets: a celebrity set and an animal set. With four distinct character configurations across all locations and scenes, the dataset comprises 404 prompts, providing a structured and diverse benchmark for evaluating story generation models. A detailed description of the dataset can be found in Appendix A.7. **Baselines.** In the existing literature, no method can generate story-driven videos solely from text, making ours the first to do so. To comprehensively evaluate the effectiveness of our method, we compare it with the corresponding Mochi backbone. We use the same generation hyperparameters to keep baseline comparisons consistent. We also compare our method with a SOTA story generation method, Vlogger Zhuang et al. (2024), adapted for our problem.

## 4.3 QUALITATIVE RESULTS

Focusing on temporal coherence, spatial consistency, and smooth transitions across video segments while preserving the semantic fidelity of input text prompts, Figures 3 illustrate results from our model along with comparisons. All frame comparisons for the videos are provided based on the original resolution. Each sequence is part of a larger generated story consisting of 12 to 13 video segments, from which we have selected key excerpts for comparison. Each sequence consists of multiple video segments $V_{12}, V_{34}, V_{56}, \ldots$ generated from discrete prompts $(P_1, P_2), (P_3, P_4), (P_5, P_6), \ldots$. Comparisons reveal that Mochi generates independent short clips without explicit transitions and has abrupt scene changes. Vlogger captures motion patterns but struggles with maintaining long-term consistency and coherent transitions. These results highlight the key advantages of our approach. Our method maintains temporal consistency by leveraging time-weighted blending, which ensures motion continuity, prevents sudden appearance changes, and preserves spatial stability by maintaining object structure, scene composition, and character placement. Semantic coherence is achieved through prompt mixing techniques that enhance alignment between generated frames and textual descriptions. Additionally, spatial-attention refinement

(SAR) plays a crucial role in enhancing local consistency by refining fine-grained details, reducing spatial artifacts, and ensuring that character features, backgrounds, and object interactions remain visually stable across frames. Overall, our approach significantly outperforms existing baselines by providing a structured, seamless, and coherent long-form video synthesis framework. More details about qualitative results are available in the Appendix A.6.

## 4.4 QUANTITATIVE EVALUATION

We evaluate the quality of generated videos using multiple metrics to measure text alignment, visual fidelity, and story consistency. CLIP Score is employed in two variants to assess text alignment: CLIP-combined, which measures overall alignment by comparing the generated frame against a composite representation of all individual text prompts, and CLIP-add, which computes the average CLIP score across individual text prompts to capture alignment with specific concepts. Higher CLIP scores indicate better text-video consistency. Additionally, the BLIP Score evaluates text-level alignment by comparing the generated frame against a combined representation of all text prompts, ensuring that the generated content aligns well with the intended textual descriptions. For visual and structural consistency, we employ DINO to measure frame-level semantic similarity, and the Learned Perceptual Image Patch Similarity (LPIPS) metric to evaluate perceptual continuity across segments. We compute the average LPIPS scores across all consecutive video segments, specifically from $V_{12}$ to $V_{(2N-1)(2N)}$, to quantify the extent to which our method maintains temporal and spatial continuity throughout the generated story. Lower LPIPS values indicate smoother transitions between segments. Considering the evaluation methodology of VideoDirectorGPT Lin et al. (2023) and the dataset scale used in the DreamRunner Wang et al. (2024b) (DreamStorySet, which consists of 13 stories with 5 to 8 scenes each) study, we use 25% of the overall data (8 stories from a total of 32 stories) from our dataset as the validation set to compute the quantitative metrics. Table 1 presents the quantitative comparison of our method against baseline approaches, including Mochi and Vlogger. Our approach achieves superior text alignment (higher CLIP scores), stronger frame-level semantic coherence (higher DINO), and enhanced story consistency (lower LPIPS), demonstrating its effectiveness in generating long-form coherent video sequences.

## 4.5 ABLATION STUDY

To analyze the impact of Time-Weighted Blending (TWB), Dynamics-Informed Prompt Weighting (DIPW), and Semantic Action Representation (SAR) on long-form video synthesis, we conduct an ablation study. Figure 4 (in Appendix) presents qualitative comparisons and Table 1 presents quantitative comparisons, illustrating the role of each component in ensuring scene consistency, structured motion transitions, and logical character interactions. The Full Model (Ours) achieves the most visually and semantically coherent storytelling, ensuring that Tom Cruise and Taylor Swift remain consistent across frames while maintaining smooth motion transitions and logical action sequences. In contrast, removing individual components leads to significant disruptions in scene consistency, prompt adherence, and motion continuity. More details (including detailed explanations and side-by-side visual comparisons) are available in the supplementary video and the Appendix A.5.

## 5 CONCLUSION

In this work, we introduced a novel framework for long-form video generation using discrete prompts by leveraging latent-space blending, time-weighted blending, and temporal attention-based prompt mixing. By incorporating temporally-aware, dynamics-inspired mechanisms, our method ensures coherent transitions between discrete video segments, achieving consistent, structured storytelling without requiring additional large-scale datasets or model retraining. Extensive experiments demonstrate significant improvements over existing short-form video synthesis techniques, bridging the gap between short clips and extended storytelling.

**Limitations and Future Directions:** In spite of its advantages, our current approach can be further improved. First, while latent space blending improves consistency, it is not always perfect at segment boundaries, particularly in highly complex motion scenarios. Second, we assume that the input textual descriptions contain sufficient semantic continuity, which may not always hold. Additionally, current diffusion models may struggle with highly dynamic scene transitions, requiring rapid or nonlinear motion changes. Computational overhead remains a concern for blending in real-time applications. Future work may further extend this framework to interactive video generation, where users can iteratively refine narratives based on real-time feedback, thereby significantly expanding its applicability in creative media and AI-assisted, user-guided content creation.

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

# A    SUPPLEMENTARY MATERIALS

## A.1    RESULTS - README: BEFORE WATCHING COMPRESSED VIDEOS

Please refer to the supplementary video for more results.

> **README: BEFORE WATCHING COMPRESSED VIDEOS**
>
> Dear Reviewers,
> Due to the **100MB submission limit** for ICLR 2026, **we had to significantly compress the supplementary files.** The original file size for the Main Results video was 168.6 MB, and the Comparison video was 38.8 MB. **Please understand that the compressed version of videos may introduce visible artifacts.** If permitted by the conference, **we would be happy to provide the original high-quality videos/high-resolution images as well.**
> Sincerely,
> The Authors

## A.2    ETHICS STATEMENT

> **Ethics Statement**
>
> All names, characters, and events appearing in this work are fictitious. They have no connection whatsoever to any real persons, places, buildings, products, or stories. All videos presented in the results are generated by an AI model based only on a text prompt dataset. The stories depicted in the results exist solely for the qualitative and quantitative evaluation of the generative model and are not intended to reflect or deliberately distort any specific individuals, organizations, regions, or historical events. It is explicitly stated that the generated stories do not represent or endorse any particular values, beliefs, cultural perspectives, or political positions in the real world. Users should be aware that the results of this research should not be considered factual information or used as a reference for decision-making.

## A.3    COMPUTATION TIME AND MEMORY CONSUMPTION

| Model | Step | Memory Consumption | Computation Time |
|---|---|---|---|
| Full Model | Inference | 31.2276 GB / 80.0 GB (29,781 MiB) | 880 sec |
| w/o DIPW | Inference | 31.2108 GB / 80.0 GB (29,765 MiB) | 357 sec |
| w/o TWB | Inference | 31.2234 GB / 80.0 GB (29,777 MiB) | 917 sec |
| w/o SAR | Inference | 30.5775 GB / 80.0 GB (29,161 MiB) | 1,205 sec |
| w/o ALL (Mochi) | Inference | 25.2801 GB / 80.0 GB (24,109 MiB) | 126 sec |

Table 2: **Step-wise Memory Consumption and Computation Time Analysis**. This table presents a detailed comparison of memory consumption and computation time for our model and its ablated variants during inference, evaluated on an NVIDIA H100 GPU (80GB). The "Full Model" includes all proposed components, achieving a balanced trade-off between efficiency and quality. Removing TWB slightly increases inference time, while excluding DIPW significantly reduces computation time. In contrast, removing SAR leads to a substantial increase in computation time, highlighting its role in optimization. The baseline "w/o ALL (Mochi)" configuration has the lowest memory and fastest inference time but lacks all of the benefits we mentioned in this paper.

Table 2 presents a detailed comparison of memory consumption and computation time across different configurations of our model, evaluated using an NVIDIA H100 GPU (80GB). The analysis highlights the impact of various components on inference efficiency, demonstrating how each contributes to overall computational requirements.

Our full model achieves high-quality synthesis with a memory consumption of 31.21 GB and an inference time of 869 seconds. The result demonstrates that while individual components contribute

to different aspects of computational performance, our full model strikes a balance between efficiency and performance. The ablation study confirms that SAR plays a crucial role in speeding up inference, while TWB and DIPW contribute to reducing overall computational time. Notably, the baseline model (Mochi) is the most lightweight but lacks the high-fidelity outputs achieved by the full model.

Overall, our approach effectively balances computational cost and memory efficiency, making it suitable for real-world applications where both scalability and high-quality synthesis are critical.

## A.4    DETAILED QUALITATIVE RESULTS

Left one in Figure 3 presents a qualitative comparison of a multi-character interaction sequence where Tom Cruise meets Taylor Swift at the Tidal Basin. Our model effectively preserves relative positioning and interactions between the two characters, while the baseline methods distort spatial relationships or fail to maintain interaction consistency. The background remains visually stable in our results, while competing methods introduce inconsistencies in the cherry blossom setting. Our model also captures emotional progression, with Tom Cruise transitioning from smiling to running and catching his breath, while other methods often fail to maintain facial expression consistency across frames. Furthermore, logical action continuity is evident in our results, as Tom Cruise's movement smoothly follows the pursuit-and-capture narrative, whereas baseline models frequently introduce abrupt or disjointed transitions. Also, the bottom one illustrates a sequential movement sequence where Tom Cruise walks through Washington, D.C., passing key landmarks such as the White House and the Washington Monument. Our model ensures smooth temporal continuity, where each motion naturally leads into the next, while some baselines generate erratic or inconsistent movements. Unlike other methods that frequently reset poses abruptly between frames, our approach maintains the logical impact of prior movements on subsequent actions. Scene awareness is also preserved, ensuring architectural landmarks remain stable, while competing methods often introduce background distortions or misplaced elements. The right one demonstrates an interaction between a corgi and a red ball in Central Park. Our model ensures object permanence, keeping the ball consistently positioned and preventing unnatural displacement, while other methods often fail to track the object properly. The action sequence follows a natural progression from the corgi seeing the ball, biting it, and then kicking it, whereas some baselines introduce inconsistencies by skipping intermediate actions or generating illogical motion transitions. Our model also captures realistic canine behavior, such as tail wagging and playful spinning, while other methods often produce rigid or unnatural movements.

## A.5    ABLATION STUDY

Without Time-weighted Blending (TWB), the generated frames lack temporal consistency, resulting in abrupt scene transitions where Tom Cruise and Taylor Swift appear as different identities across frames. Additionally, there is no meaningful interaction between them, making the sequence feel disconnected. The full model utilizes TWB to enforce bidirectional constraints, preserving spatial and temporal continuity across video segments. Without Dynamics-Informed Prompt Weighting (DIPW), the model fails to integrate Prompt 2 into the scene, leading to incomplete or inaccurate action sequences. The absence of DIPW prevents the model from smoothly interpolating between different prompt levels, causing a loss of intended action details and reducing narrative control. Our approach leverages DIPW to guide structured prompt blending, ensuring accurate action progression aligned with the evolving scene. Without Semantic Action Representation (SAR), the continuity of motion and action sequences deteriorates. The absence of SAR leads to disjointed character actions, where movements do not logically connect across frames, disrupting motion coherence. The full model incorporates SAR to encode high-level action semantics, ensuring that character behaviors evolve naturally and respond dynamically to preceding movements. These results highlight the necessity of each component: TWB maintains scene and identity consistency, DIPW enables structured prompt-driven action transitions, and SAR ensures smooth motion continuity and logical action sequences. The full integration of these modules allows for semantically aligned, visually coherent, and perceptually smooth long-form video generation.

Table 1 shows our evaluation results for ablation study. The full model (Ours) achieves the best overall performance, demonstrating that each module plays a crucial role in generating coherent

| Method | CLIP-add ↑ | CLIP-combined ↑ | BLIP ↑ | DINO ↑ | LPIPS $(V_{12} - V_{(2N-1)(2N)})$ ↓ |
|---|---|---|---|---|---|
| Vlogger | 0.2889 ± 0.0133 | 0.2998 ± 0.0122 | 28.8509 ± 1.2651 | 0.9254 ± 0.0188 | **0.6193** ± 0.0214 |
| Mochi | 0.2940 ± 0.0139 | 0.3103 ± 0.0180 | **29.5948** ± 1.8717 | 0.9530 ± 0.0098 | 0.6811 ± 0.0386 |
| **Ours (Full Model)** | **0.3055** ± 0.0095 | **0.3211** ± 0.0122 | 29.5117 ± 1.6253 | **0.9697** ± 0.0056 | 0.6733 ± 0.0501 |
| Ours w/o DIPW | 0.2883 ± 0.0214 | 0.3013 ± 0.0213 | 28.9450 ± 1.7390 | 0.9418 ± 0.0131 | 0.6412 ± 0.0224 |
| Ours w/o TWB | 0.2944 ± 0.0152 | 0.3030 ± 0.0157 | 29.1403 ± 1.1490 | 0.9655 ± 0.0093 | 0.6803 ± 0.0440 |
| Ours w/o SAR | 0.2926 ± 0.0107 | 0.3005 ± 0.0124 | 28.7187 ± 0.8393 | 0.9526 ± 0.0150 | **0.6396** ± 0.0324 |

Table 3: **Quantitative evaluation of generated videos (mean ± std).** Our method shows the best or comparable performance across multiple metrics. Lower LPIPS indicates better realism and temporal consistency. Standard deviations are omitted for brevity.

long-form videos. Although LPIPS is lower when SAR is removed (0.6396), this does not indicate better video quality. Instead, it reflects reduced motion complexity and less dynamic transitions, as SAR enhances character interactions and logical action continuity at the cost of slightly increased perceptual differences. The drop in CLIP-add (0.2926) and CLIP-combined (0.3005) without SAR further confirms that it is essential for maintaining text-video alignment. Similarly, removing DIPW leads to weaker prompt-based scene transitions, while TWB removal results in the highest LPIPS (0.6887), indicating degraded temporal smoothness. These results highlight that SAR, DIPW, and TWB must be combined to ensure text-aligned, semantically structured, and perceptually coherent video generation. The method for quantitative evaluation follows Section 4.4.

## A.6 Detailed Quantitative Evaluation

Table 3 reports the same quantitative evaluation as Table 1, but includes standard deviation values (mean ± std) computed across 8 validation stories. The inclusion of standard deviations provides insight into the stability and consistency of each method across diverse prompts and scenes. Our full model not only achieves the best average performance across all metrics—including CLIP-add, CLIP-combined, BLIP, DINO, and LPIPS—but also shows stable results with relatively low variance. Each ablation variant (w/o DIPW, TWB, SAR) demonstrates noticeable drops in performance or increased variability, highlighting the contribution of each component to the overall video generation quality.

## A.7 Dataset Plot and Character

Our dataset is designed to comprehensively represent both animate (e.g., humans, animals) and inanimate objects (e.g., balls, buses, cars, boats, airplanes) to ensure diversity in story generation. Each story plot includes at least one visually distinguishable action performed by an entity, such as throwing a ball, boarding a vehicle, running, pressing a button, or walking, to enhance dynamic storytelling.

The dataset covers eight diverse locations: New York City, Washington D.C., Paris, London, Los Angeles, San Francisco, Chicago, and Las Vegas. Among these, seven locations (excluding New York City) feature at least two characters (or two animals) per plot, introducing interactions and multi-character dynamics. Each story plot consists of 12 to 13 sequential scenes, with Chicago, Las Vegas, and New York City including 12 scenes per story, while all other locations include 13 scenes per story. To ensure diversity in character representation, we introduce two distinct character sets: a celebrity set (Tom Cruise & Taylor Swift, Elon Musk & Angelina Jolie) and an animal set (Corgi Dog & Siamese Cat, Panda & Fox). Given eight different locations, 12-13 scenes per story, and four distinct character settings, the dataset consists of a total of 404 prompts. ($12 \times 3 + 13 \times 5 = 101$, $101 \times 4 = 404$)

Prompt 1 provides a broad scene description that establishes the setting and context, while Prompt 2 introduces specific actions performed by the character within that scene. For example, in a New York City subway scenario, Prompt 1 may be "Tom Cruise is inside of the subway train," setting up the environment, whereas Prompt 2 specifies an action such as "Tom Cruise is sitting." This structure enables fine-grained control over character movement and interactions while maintaining coherence in scene transitions.

This structured approach allows for a broad range of environments, interactions, and character-driven narratives, making it well-suited for evaluating story generation models. Actions that humans can perform but animals cannot may be adapted accordingly. For example, since a dog cannot pick up and throw a ball with both hands, such an action is replaced with the dog kicking the ball instead.

To be specific, in a scene where the character arrives at Central Park, Prompt 1 describes, "Tom Cruise sees a red ball in Central Park," providing situational context, while Prompt 2 refines the action in detail with, "Tom Cruise is picking up a red ball in Central Park" and later "Tom Cruise is throwing a red ball in Central Park." Similarly, in an animal-based variation, Prompt 1 states, "A corgi dog sees a red ball in Central Park," while Prompt 2 adapts the action appropriately, such as "A corgi dog is biting a red ball in Central Park" and "A corgi dog is kicking a red ball in Central Park."

This approach ensures that actions are naturally adapted for different entities, particularly when an action performed by a human (e.g., gripping and throwing a ball) must be substituted with a more plausible behavior for an animal (e.g., biting or kicking the ball). The dataset maintains consistency in narrative progression while allowing for variations based on both character type and setting. Some examples of dataset plots can be found below.

```
prompt_nyc = [
    "Tom Cruise is inside of the subway train", "Tom Cruise is sitting",
    "Tom Cruise is looking out the subway window", "Tom Cruise now stands
        out",
    "Tom Cruise is getting off the NYC subway train", "Tom Cruise is
        walking",
    "Tom Cruise is walking up the subway exit stairs", "Tom Cruise is
        looking around",
    "Tom Cruise is looking at the streets of Times Square, NYC", "Tom
        Cruise is tilting his head curiously",
    "Tom Cruise is walking on the streets of Times Square, NYC", "Tom
        Cruise is walking",
    "Tom Cruise is waiting for a bus at the Times Square bus stop in NYC
        ", "Tom Cruise is standing",
    "Tom Cruise is getting on a bus at the Times Square bus stop", "Tom
        Cruise is walking",
    "Tom Cruise is looking out the bus window at the city view", "Tom
        Cruise is sitting",
    "Tom Cruise has arrived at Central Park", "Tom Cruise is strolling",
    "Tom Cruise sees a red ball in Central Park", "Tom Cruise is
        observing it curiously",
    "Tom Cruise is picking up a red ball in Central Park", "Tom Cruise is
         gripping it firmly",
    "Tom Cruise is throwing a red ball in Central Park", "Tom Cruise is
        watching its trajectory"
]
prompt_nyc_corgi = [
    "a corgi dog is inside of the subway train", "a corgi dog is sitting
        ",
    "a corgi dog is looking out the subway window", "a corgi dog now
        stands out",
    "a corgi dog is getting off the subway train", "a corgi dog is
        walking",
    "a corgi dog is looking at the streets of Times Square, NYC", "a
        corgi dog is tilting its head curiously",
    "a corgi dog is walking on the streets of Times Square, NYC", "a
        corgi dog is wagging its tail",
    "a corgi dog is waiting for a bus at the Times Square bus stop in NYC
        ", "a corgi dog is standing",
    "a corgi dog is getting on a bus at the Times Square bus stop", "a
        corgi dog is walking",
    "a corgi dog is looking out the bus window at the city view", "a
        corgi dog is sitting",
    "a corgi dog has arrived at Central Park", "a corgi dog is sniffing
        the ground",
```

```
      "a corgi dog sees a red ball in Central Park", "a corgi dog is
          looking up at the sky",
      "a corgi dog is biting a red ball in Central Park", "a corgi dog is
          wagging its tail",
      "a corgi dog is kicking a red ball in Central Park", "a corgi dog is
          playfully spinning"
]
prompt_dc = [
      "Tom Cruise is walking around the White House", "Tom Cruise is
          observing the architecture",
      "Tom Cruise is strolling along the streets near the U.S. Capitol in
          Washington D.C.", "Tom Cruise is looking around with curiosity",
      "Tom Cruise is passing by the Washington Monument", "Tom Cruise is
          walking",
      "Tom Cruise is walking along the Tidal Basin surrounded by cherry
          blossoms", "Tom Cruise is taking a leisurely stroll",
      "Tom Cruise stops for a moment at the Tidal Basin surrounded by
          cherry blossoms", "Tom Cruise is sitting",
      "Tom Cruise is jogging along the Tidal Basin surrounded by cherry
          blossoms", "Tom Cruise is enjoying the fresh air",
      "Tom Cruise encounters Taylor Swift at the Tidal Basin surrounded by
          cherry blossoms", "Tom Cruise is smiling",
      "Tom Cruise is trying to catch Taylor Swift at the Tidal Basin
          surrounded by cherry blossoms", "Tom Cruise is running",
      "Taylor Swift is eventually caught up with by Tom Cruise at the Tidal
           Basin surrounded by cherry blossoms", "Tom Cruise is catching
          his breath and grinning",
      "Tom Cruise and Taylor Swift are enjoying the cherry blossoms
          together", "Tom Cruise and Taylor Swift are sitting side by side
          ",
      "Tom Cruise and Taylor Swift are admiring the cherry blossoms at the
          Tidal Basin", "Tom Cruise and Taylor Swift are lying on the grass
           looking up at the sky",
      "Tom Cruise and Taylor Swift are lying on the grass at the Tidal
          Basin surrounded by cherry blossoms, slowly closing their eyes",
          "Tom Cruise and Taylor Swift are resting peacefully",
      "Tom Cruise and Taylor Swift have fallen asleep at the Tidal Basin
          surrounded by cherry blossoms", "Tom Cruise and Taylor Swift are
          peacefully dozing off"
]
prompt_dc_corgi = [
      "a corgi dog is walking around the White House", "a corgi dog is
          sniffing the ground",
      "a corgi dog is strolling along the streets near the U.S. Capitol in
          Washington D.C.", "a corgi dog is walking",
      "a corgi dog is passing by the Washington Monument", "a corgi dog is
          looking around",
      "a corgi dog is walking along the Tidal Basin surrounded by cherry
          blossoms", "a corgi dog is taking a leisurely stroll",
      "a corgi dog stops for a moment at the Tidal Basin surrounded by
          cherry blossoms", "a corgi dog is sitting",
      "a corgi dog is running along the Tidal Basin surrounded by cherry
          blossoms", "a corgi dog is excitedly running",
      "a corgi dog encounters a siamese cat at the Tidal Basin surrounded
          by cherry blossoms", "a corgi dog is tilting its head curiously",
      "a corgi dog is chasing a fleeing siamese cat with at the Tidal Basin
           surrounded by cherry blossoms", "a corgi dog and a siamese cat
          are running",
      "a siamese cat is finally caught by a corgi dog at the Tidal Basin
          surrounded by cherry blossoms", "a corgi dog is gently wagging
          its tail",
      "a corgi dog and a siamese cat are sniffing the scent of cherry
          blossoms together at the Tidal Basin", "a corgi dog and a siamese
           cat are sitting",
```

```
    "a corgi dog and a siamese cat are enjoying the cherry blossoms at
        the Tidal Basin", "a corgi dog and a siamese cat are lying down",
    "a corgi dog and a siamese cat are lying down at the Tidal Basin
        surrounded by cherry blossoms, slowly closing their eyes", "a
        corgi dog and a siamese cat are lying down",
    "a corgi dog and a siamese cat have fallen asleep at the Tidal Basin
        surrounded by cherry blossoms", "a corgi dog and a siamese cat
        are peacefully sleeping"
]
(...)
```

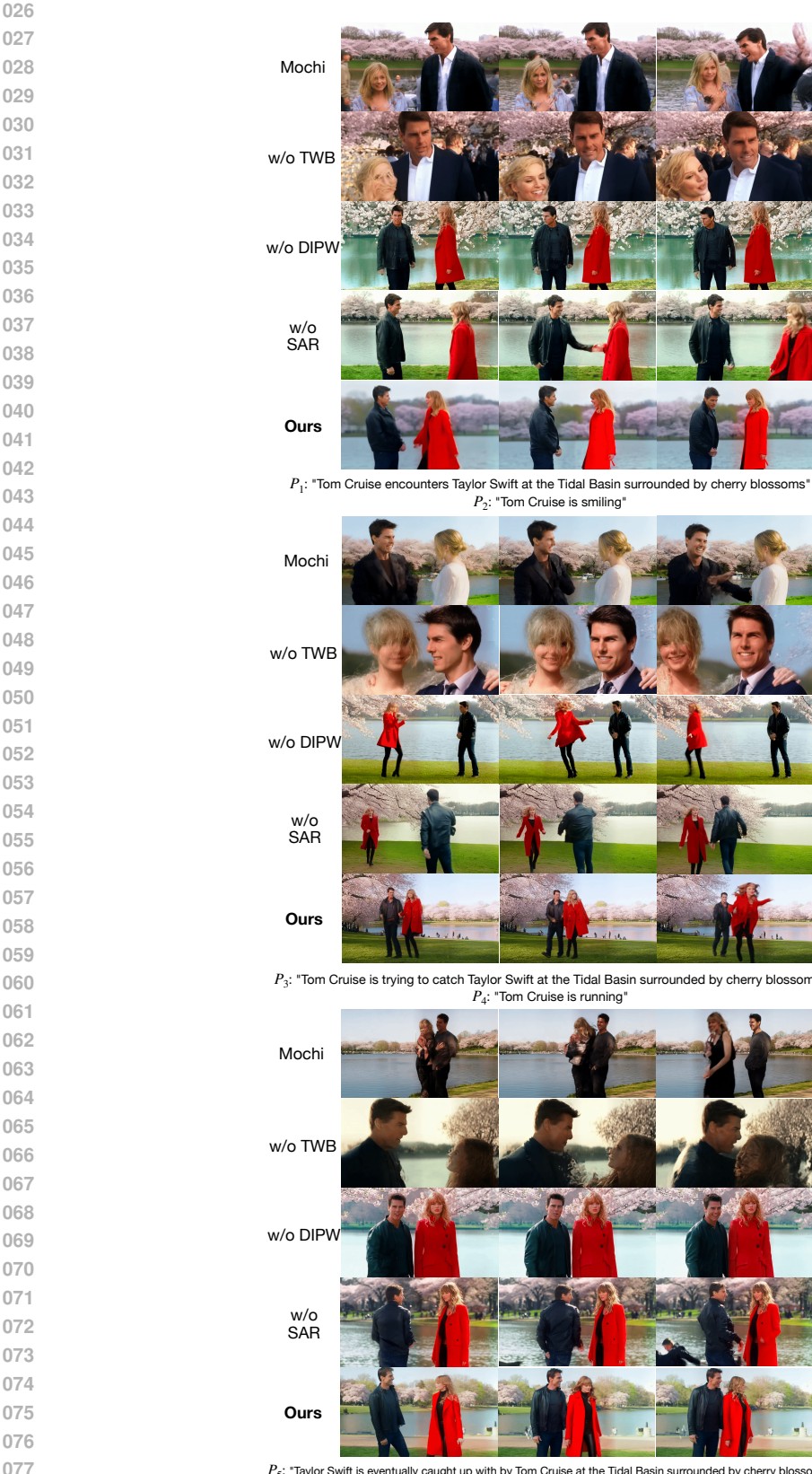

Figure 4: **Ablation Study** Each row shows a different setting: Full Model, w/o TWB, w/o DIPW, w/o SAR, and a baseline (Mochi). The full model produces the most coherent motion and character interactions.

