# OpenReview forum: "Dynamics-Inspired Text-Guided Video Storytelling"
_ICLR.cc/2026/Conference — ICLR 2026 Conference Withdrawn Submission_

### Official Review · Reviewer_eX6B · 2025-10-29

**Soundness:** 3
**Presentation:** 2
**Contribution:** 2
**Rating:** 4
**Confidence:** 4

**Summary:**

This paper proposes a novel method for storytelling video generation that can do seamless transitions between generated video segments. The main contributions are three aspects: dynamics-informed prompt weighting, temporally-aware blending with bidirectional constraints, and structured semantic action representation. Dynamics-informed prompt weighting can balance the contributions of two prompts. The second contribution designs a time-weighted blending mechanism that dynamically balances past and future frames. The third contribution encodes high-level action semantics into the blending process using a pre-trained text encoder.

**Strengths:**

1. The paper is well organized.

2. The topic of storytelling long video generation is worth exploring in the research community. This paper targets this important problem.

3. The paper provides video demonstrations to help reviewers better evaluate the performance of the proposed method.

**Weaknesses:**

There are some concerns and questions about this paper:

1.	The description of the Dynamics-Informed Prompt Weighting method seems complicated. Could the author provide diagrams to help readers understand it?

2.	According to Formula 3, the generation of the next video clip depends on the generation of the previous video clip. Won't such an operation introduce error accumulation?

3.	Where is alpha^’ applied in Formula 7? How exactly is it used?

4.	The long video demo shown in the supplementary materials looks more like a series of video clips spliced together, with very obvious transitions between each video clip, making it difficult for viewers to have an immersive experience.

5.	With only two comparison methods, could the authors compare it with more methods? For example, One-Minute Video Generation with Test-Time Training (CVPR 2025, open-source).

6.   The three innovations proposed in the paper lack a clear connection. For example, when introducing SAR, the authors only mention that "Preserving temporal smoothness alone is insufficient," but why? Is this problem caused by the introduction of the first two innovations? It would be helpful if the authors could more clearly explain the connection and motivation between each innovation.

**Questions:**

Please see above. I think the authors should compare with more baseline methods. In addition, the video demo shown in the supplementary material is not very impressive.

---

### Official Review · Reviewer_5TEx · 2025-10-30

**Soundness:** 2
**Presentation:** 3
**Contribution:** 2
**Rating:** 4
**Confidence:** 3

**Summary:**

This paper presents a novel, training-free framework to generate coherent, long-form videos from discrete text prompts, addressing the common issues of temporal and semantic inconsistency in existing text-to-video models. The core contribution is a three-part system that integrates Dynamics-Informed Prompt Weighting (DIPW) to balance scene and action cues, Time-Weighted Blending (TWB) for smooth transitions between video segments, and a Semantic Action Representation (SAR) to ensure logical motion continuity. By combining these dynamics-inspired mechanisms, the method enforces consistency across multiple short video clips to form a cohesive narrative. Extensive experiments show the proposed approach significantly outperforms baselines in generating temporally consistent and visually compelling video stories.

**Strengths:**

1. The authors propose a training-free solution to address the consistency challenge in long-form video generation. I believe the proposed Dynamics-Informed Prompt Weighting (DIPW), Time-Weighted Blending (TWB), and Semantic Action Representation (SAR) modules are effective and beneficial to the community.

2. The dual-prompt formulation that decouples the textual description into scene-level and action-level is an interesting and insightful design.

3. A new benchmark is proposed to facilitate the following research. And the experimental results show that the proposed method outperforms several baseline methods in the proposed benchmark.

**Weaknesses:**

1. I do not think the proposed framework can process the storytelling task that includes apparent semantic change. The TWB module computes the new initial states of the next segment as the blended latent embeddings of the former segments, yet how does it handle the situation when there is an intentional, abrupt transition between two consecutive segments? The SAR module tackles the semantically different segments, yet I am concerned about its performance since it only uses the CLIP embeddings as the distinguisher. Can it reliably distinguish between a continuation of a similar movement and a completely new movement? In the qualitative demos, consecutive video segments are highly semantically similar, so I am curious about how the model would perform when the storytelling requires more variable visual content.

2. The entire framework is based on Mochi, yet almost all ablation variants perform worse the the original Mochi version. What is the reason? I believe these three modules are independent. If each of them is helpful, it will boost the performance of the vanilla Mochi version. Besides, on some metrics, the full model shows less competitive results compared to Mochi. Why does the performance become worse after adapting these modules?

3. The applicability of the proposed method may be highly restricted by its inference speed. As the authors complement in Appendix A.3, the average inference speed of the proposed method is around 7x times that of the original Mochi method.

**Questions:**

1. The entire framework is based on Mochi. Can the proposed method be applied to other storytelling or T2V/I2V generation methods, such as Vlogger? Besides, how does the proposed method rely on the overall performance of the pretrained Mochi model? Will the proposed method correct the failure cases of Mochi?

2. In the DIPW module, how sensitive is the video quality to the fixed weight hyperparameters and temperature?

3. Formulas on page 5 do not have formula numbers, and the numbers start on page 6.

---

### Official Review · Reviewer_MAwd · 2025-10-31

**Soundness:** 2
**Presentation:** 2
**Contribution:** 2
**Rating:** 2
**Confidence:** 4

**Summary:**

This paper focuses on the challenges of maintaining consistency and action continuity in the task of Video Storytelling. The authors propose a series of methods, including Time-weighted Blending, which preserves historical information; Dynamics-Informed Prompt Weighting, which adjusts the influence ratio of the double-prompt mechanism; and Semantic Action Representation. The proposed approach demonstrates promising results on the Video Storytelling task, enabling the generation of multi-scene long-form videos while effectively maintaining both subject consistency and action continuity.

**Strengths:**

- The paper focuses on the long-form video storytelling task and highlights the challenges of subject consistency and action continuity, which remain pressing issues in the field.
- The proposed Time-weighted Blending mechanism offers a novel perspective on maintaining consistency in long-form video generation.

**Weaknesses:**

- **Ambiguous problem definition**
    - The Abstract states that the paper focuses on long-form storytelling while also mentioning the goal of preventing abrupt transitions, which makes the problem formulation somewhat unclear. It is not entirely evident whether the focus is on pixel-level scene transitions or multi-shot video generation.
    - The authors identify temporal coherence, semantic meaning, and action continuity as the main challenges. However, if I understand correctly, semantic meaning and action continuity can be considered components of temporal coherence. Since the work primarily aims to ensure consistency across multiple video segments, it might be helpful for the authors to clarify whether temporal coherence refers to high-level semantic consistency rather than pixel-level smoothness. A clearer statement of the task definition and key challenges would strengthen the paper’s focus.
- **Lack of clarity in presentation**
    - The Abstract and Introduction introduce specific module names (e.g., Time-weighted Blending, Dynamics-Informed Prompt Weighting) without prior explanation, which may make it difficult for readers to follow at first.
    - Section 3 would benefit from more detailed descriptions. For example:
        - It is unclear whether the generated frame refers to historical or current frames (L240).
        - The difference between the subscripts $T$ in Eq. 3 and $N−1$ in Eq. 1 is not well explained.
        - The hyperparameter $\gamma$ in Eq. 4 is not specified.
        - The role of $\alpha'$ in Section 3.3 and the motivation behind the design of the SAR mechanism are not clearly described. Providing further clarification on these aspects would make the method section easier to follow.
- **Limited evaluation scope**
    - For the task of maintaining consistency across multiple video segments, it would be beneficial to include comparisons with additional baselines such as reference-to-video methods [1] and multi-shot video generation models [2–3].
    - The evaluation currently relies mainly on DINO and LPIPS, which capture only part of the intended objectives. The authors are encouraged to include more comprehensive metrics, such as background consistency (as in VBench [4]) and ViCLIP feature similarity, to provide a fuller assessment of video consistency and continuity.
- **Limited qualitative diversity**
    - The qualitative results mostly feature similar subject pairs (e.g., A Woman & A Man, Dog & Cat). Including a wider range of examples would better demonstrate the generalization and robustness of the proposed method.

[1] Liu L, Ma T, Li B, et al. Phantom: Subject-consistent video generation via cross-modal alignment[J]. arXiv preprint arXiv:2502.11079, 2025.(ICCV 2025)

[2] Long F, Qiu Z, Yao T, et al. Videostudio: Generating consistent-content and multi-scene videos[C]//European Conference on Computer Vision. Cham: Springer Nature Switzerland, 2024: 468-485. (ECCV 2024)

[3] Zhou Y, Zhou D, Cheng M M, et al. Storydiffusion: Consistent self-attention for long-range image and video generation[J]. Advances in Neural Information Processing Systems, 2024, 37: 110315-110340. (NIPS 2024)

[4] Huang Z, He Y, Yu J, et al. Vbench: Comprehensive benchmark suite for video generative models[C]//Proceedings of the IEEE/CVF Conference on Computer Vision and Pattern Recognition. 2024: 21807-21818.

**Questions:**

I have several questions regarding the task definition, design choices, and evaluation results, which may help clarify the authors’ approach.
- How do the authors define the scope of the Video Storytelling task? Based on the examples provided in the supplementary material, the notion of consistency appears to focus mainly on subject and action. Could the authors clarify whether scene consistency is also considered within the task definition?
- Regarding the examples involving multiple character sequences, maintaining consistency across multiple subjects may introduce additional complexity compared to single-subject scenarios. A brief discussion on whether multi-subject consistency poses distinct challenges could strengthen the presentation.
- The proposed approach introduces discrete prompts that separately describe the main content and actions. Have the authors explored using more detailed or multi-aspect prompts, e.g., generating multiple samples with enriched textual descriptions and then concatenating them into a long video, for comparison? How different do the authors expect such detailed prompts to be from the proposed discrete prompt formulation?
- In Table 1, the proposed method performs lower on LPIPS compared with other methods. What might be the reason for this? Additionally, when action representations are removed, the LPIPS score improves. Could the authors provide insights into why this might occur?

---

### Official Review · Reviewer_Udbq · 2025-11-01

**Soundness:** 2
**Presentation:** 4
**Contribution:** 2
**Rating:** 4
**Confidence:** 3

**Summary:**

This paper proposes a storytelling framework for long-form video generation, enabling coherent video sequences. The method achieves this through a DIPW mechanism that adaptively balances scene and action prompts at each diffusion timestep, and a TWB strategy to ensure temporal consistency between video segments. Additionally, a SAR is incorporated to enhance motion continuity, allowing smooth and adaptive transitions. The experiments demonstrate is able to produce long-form videos.

**Strengths:**

1. The paper has a clear structural logic and a well-organized chapter arrangement, which makes it easy for readers to understand the method and the experimental.
2. The proposed method can be directly applied without requiring additional training data or fine-tuning.
3. The paper achieves long-form video generation by designing dedicated metrics to balance the weights of action and background prompts while considering adjacent frames to achieve smooth transitions.

**Weaknesses:**

1. The proposed method involves many hyperparameters, but the paper currently lacks sufficient visualizations or ablation studies to demonstrate the stability of the method and its sensitivity to hyperparameters. I suggest the authors provide additional experiments or analyses to verify the robustness of the method under different hyperparameter settings.
2. In DIPW, the similarity between generated frames and text prompts is computed at each denoising timestep. However, in the early stages of denoising, the generated frames often contain substantial noise, and CLIP’s performance on noisy images is not reliable. This may further affect the accuracy of the prompt scores.
3. It is not clearly explained how SAR uses the blending factor α’ to control the Video Diffusion model to enhance motion continuity and ensure smooth transitions.

**Questions:**

1. Based on the videos provided in the attachment, Mochi, which shares the same base model, still noticeably lags behind the proposed method in fundamental video generation capabilities, such as visual fidelity and temporal consistency. Considering that the method is primarily optimized for storytelling capabilities, why does it also show significant improvements in the basic video generation abilities?

2. Regarding Equation 1, why is the frame index i of $z_{(2N-3)(2N-2),i}$ set from 0 to N−1 instead of from 0 to T?

---

### Note · Authors · 2025-11-12

I have read and agree with the venue's withdrawal policy on behalf of myself and my co-authors.